# Effective Treatment of Traumatic Brain Injury in Rowett Nude Rats with Stromal Vascular Fraction Transplantation

**DOI:** 10.3390/brainsci8060112

**Published:** 2018-06-18

**Authors:** Sean Berman, Toni L. Uhlendorf, Mark Berman, Elliot B. Lander

**Affiliations:** 1California Stem Cell Treatment Center, Beverly Hills, CA 90201, USA; mark@cellsurgicalnetwork.com (M.B.); elliot@cellsurgicalnetwork.com (E.B.L.); 2Department of Biology, California State University, Northridge, CA 91330, USA; Toni.Uhlendorf@csun.edu

**Keywords:** Traumatic brain injury (TBI), concussion, stromal vascular fraction (SVF), adipose derived stem cells (ADSCs)

## Abstract

Traumatic brain injury (TBI) affects 1.9 million Americans, including blast TBI that is the signature injury of the Iraq and Afghanistan wars. Our project investigated whether stromal vascular fraction (SVF) can assist in post-TBI recovery. We utilized strong acoustic waves (5.0 bar) to induce TBI in the cortex of adult Rowett Nude (RNU) rats. One hour post-TBI, harvested human SVF (500,000 cells suspended in 0.5 mL lactated Ringers) was incubated with Q-Tracker cell label and administered into tail veins of RNU rats. For comparison, we utilized rats that received SVF 72 h post-TBI, and a control group that received lactated Ringers solution. Rotarod and water maze assays were used to monitor motor coordination and spatial memories. Rats treated immediately after TBI showed no signs of motor skills and memory regression. SVF treatment 72 h post-TBI enabled the rats maintain their motor skills, while controls treated with lactated Ringers were 25% worse statistically in both assays. Histological analysis showed the presence of Q-dot labeled human cells near the infarct in both SVF treatment groups; however, labeled cells were twice as numerous in the one hour group. Our study suggests that immediate treatment with SVF would serve as potential therapeutic agents in TBI.

## 1. Introduction

Traumatic brain injury (TBI) is a severe injury that affects up to 3.8 million Americans each year with many more TBI incidents going unreported [1]. TBI occurs when a blow to the head causes the brain to be displaced beyond the blood brain barrier causing a subdural hematoma and damaging neuronal cells [2]. When the brain is displaced beyond the cerebrospinal fluid it sits in, it crashes into the skull, causing blood vessels in the brain to sheer and break [3]. Blast waves, resulting from overpressures created by explosive devices, can also cause a TBI due to rapid and extreme changes in atmospheric pressure. These changes can cause tissues, like those in the brain, to be displaced, stretched and sheered, resulting in a blast TBI [4]. This disruption in the cerebral vasculature prevents adequate blood flow to neuronal cells, which can eventually become depleted of oxygen, nutrients and eventually die via necrosis and apoptosis [5,6]. Neuronal injury and death are directly associated with the short-term symptomatic effects (dizziness, severe headaches and complete loss of consciousness) and the long-term effects (memory loss, depression and suicidality) [7].

The ability to diagnose concussions has greatly increased in recent years, however there is presently little to actively treat the diagnosed TBI [8]. Currently, individuals are directed to rest until symptoms dissipate or go away completely and prescribed analgesics to mitigate pain [8]. Due to the nature of a TBI in which cellular damage occurs, the immune system is upregulated and permeability of the blood brain barrier (BBB) is induced, mesenchymal stem cells present a viable innovative treatment option [9]. This study looks to administer stromal vascular fraction (SVF) intravenously to actively treat TBI in rats.

SVF is a mixture of cells consisting of stromal cells, pericytes, endothelial cells, endothelial progenitor cells, leukocytes, and adipose derived stem cells (ADSCs) [10]. SVF can be easily obtained through a minimally invasive liposuction aspiration procedure. By definition, ADSCs found within SVF are multipotent and can thus replicate and differentiate into bone, cartilage, adipose tissue, muscle, vascular tissue and neural tissue [11]. ADSCs are also attracted to sites of inflammation throughout the body [12]. The combination of ADSC’s multipotency, migratory ability to sites of inflammation and ease of accessibility, make them an attractive treatment option for TBI, which without an invasive procedure (craniotomy or arterial injection) is otherwise inaccessible for treatment [11,12].

This study looks to actively treat TBI with SVF in a minimally invasive way. The aim here is to induce TBI in an animal model, and discover how SVF treatment can be most efficacious, mimicking clinical therapies in humans. The current study employs the use of an acoustic wave technology to inflict the TBI without fracturing the skull. By utilizing the shockwave technology of the Stroz-D-Actor, acoustic shockwaves penetrate the bony skull and mimic the blast TBI often seen in military combat [13]. The study also looks to inject SVF at two different time periods intravenously to gauge the window of opportunity available for potential treatment for the mitigation of induced TBI.

## 2. Materials and Methods

### 2.1. Rowett Nude Rat (RNU)

This experiment was performed using 95–100 day old male Rowett Nude immunosuppressed rats (RNUs). Rats were taken from an RNU rat colony maintained at California State University Northridge. Three RNU groups (*n* = 18 total) were tested in this experiment. All animals received a TBI on day 0. Immediate Group (*n* = 6) received a 0.5 mL tail vein injection of SVF shortly after TBI on day 0. 72 Hour Group (*n* = 6) received a 0.5 mL tail vein injection of SVF at 3 days post-TBI. Control Group (*n* = 6) received a 0.5 mL tail vein injection of only lactated Ringer’s solution on day 0 immediately post-TBI. All animals underwent the same behavioral testing on 3 nonconsecutive days, beginning 5 days prior to TBI. All protocols utilized in this study were approved by California State University, Northridge’s Institutional Animal Care and Use Committee (IACUC).

### 2.2. TBI Induction

The Storz-D-Actor, commonly used in orthopedic, urology, cardiology, and aesthetic medical practices, was used to induce a closed head blast TBI. The device emits an acoustic wave which penetrates the skull and causes neural damage similar to that of a mild to moderate TBI [13].

Each RNU rat was anesthetized with 2.5% isoflurane mixed in O_2_ for 45 s before receiving a TBI. Each RNU rat received a single pulse of acoustic wave energy that was emitted at 5.0 bar from the Storz-D-Actor. The pneumatic hand-held device was held constantly posterior to the rat’s left eye so that the acoustic wave was directed to the left frontal motor cortex of the brain. Immediately after the TBI was induced, rats were rolled onto their backs, and the time it took to right themselves was recorded.

### 2.3. Stromal Vascular Fraction Treatment

The SVF for this research was obtained from the Cell Surgical Network that received consent from anonymous patients who had been previously cleared of any blood borne disease (ICSS-2016-018). To extract the SVF, patients received local sub-dermal anesthetic, and a mini-liposuction was performed. After extraction of the adipose tissue, the cells were incubated in a closed sterile container with GMP grade collagenase (Roche, Indianapolis, IN, USA,) for enzymatic digestion of the constricting extracellular matrix. A Time Machine™ centrifuge (Medikan International, Kangnam, South Korea) to isolate the SVF in accordance with Cell Surgical Network standard protocol.

The cells in the SVF were first counted using the Invitrogen Cell Countess I (Waltham, MA, USA) to obtain an estimated count prior to flow cytometry analysis. Hematopoietic stem cells (HSCs) were identified by CD45^+^, DC34^low^, CD14^+^, and CD31^-^ cell surface markers. Adipose derived stem cells (ASCs) were identified by CD45^-^, CD34^+^ and CD90^+^ cell surface markers [14]. After the SVF was harvested, the cells were incubated with Qtracker 625 cell labeling materials in RPMI media on the same day. Once the cells were prepared with Q-Dots and incubated for 2 h, approximately 500,000 stromal vascular fraction cells were suspended in 0.5 mL of lactated Ringer’s solution and placed into 1 mL sterile syringes. The freshly harvested and isolated SVF was delivered via tail vein injection to all TBI experimental animals within a 5 h window. Negative control TBI rats received lactated Ringer’s via tail vein injection at the same time as their experimental pairs.

### 2.4. Rotarod Test

To test for motor coordination impairment, all animals underwent three nonconsecutive days of rotarod training, five days prior to the administration of TBI. The animals were trained on the rotarod (Med Associates Inc., St. Albans, VT, USA), which increased in speed from 4 to 40 rpm over the course of 300 s. On training days (including one hour prior to TBI induction) and testing days (post-TBI), the animals performed the test three times, and their rotarod results were averaged. On the final day of training (day 0), the animals were left on the rotarod as long as possible to record their maximum performance capability or which was used as baseline to compare post-TBI performance. Their post-implant, rotarod performances were tested on 1, 4, 7, and 14 days post-TBI. Each animal’s rotarod latency times were compared to their individual baseline performance and used to determine percentage of peak performance.

### 2.5. Morris Water Maze Test

To test for memory impairment, we utilized a water maze. A round black plastic tub (one meter diameter) was divided into four quadrants and filled (20 cm) with tap water. The water (18 ± 1 °C) was made opaque by using dehydrated milk powder. Above the water line, each quadrant was marked with unique, orienting white tape. A glass platform (5 cm diameter) was always placed in the same location, 2.5 cm below the waterline in the 3rd quadrant. The rats were placed in the water facing away from the platform in the 1st quadrant. The time (swim latency) it took for the rat to turn around, swim to the submerged platform and stand on top of it for at least a full second was recorded. Similar to the rotarod test, the animals underwent three nonconsecutive days of training in the water maze, starting five days prior to TBI induction. Three daily trials were completed with five minutes of rest in between each trial. Each water maze test was conducted after completion of rotarod testing. After TBI induction, animals were retested on days 1, 4, 7, and 14. Their swim latency times (seconds) were recorded for three nonconsecutive trials and averaged. Each animal’s swim latency times were determined as a percentage of their baseline performance (taken one hour prior to TBI administration).

### 2.6. Histology

Fourteen days post-TBI administration after the final behavioral assays were run, all animals were injected IP with 400 mg/kg chloral hydrate. The animals were transcardially perfused and fixed with 4% paraformaldehyde dissolved in 0.1 M phosphate buffered solution (PBS). Their brains were removed, and post-fixed in 4% paraformaldehyde/ 0.1 M PBS for 48 h at 4 °C. The fixed brains were then placed in a solution of 20% sucrose in paraformaldehyde/0.1 M PBS for an additional 48 h (also at 4 °C) prior to sectioning. The frontal cortex was sliced on a cryostat on the coronal plane (25 μm thick) at the same stereological level for all animals. The tissue was arranged on a glass slide for drying, and then stained with cresyl violet.

### 2.7. Statistics

Statistical analyses were performed on data collections. Multiple comparisons of data collected from rotarod and water maze assays were analyzed by repeated measures ANOVA. All figures are displayed with means ± SE bars.

## 3. Results

### 3.1. SVF Characterization of Stem Cells

The SVF used in this project was analyzed by flow cytometry to determine the amount of adipose derived stem cells (ADSC) and hematopoietic stem cells (HSC) found within our sample. ADSCs were characterized by the presence of CD45^-^, CD34^+^, and CD90^+^ markers. HSCs were characterized by the presence of CD45^+^, CD34^low^, CD14^+^, and CD31^-^ markers. The SVF sample used contained 11,354 ADSCs per mL (8.3% of total cells) and 10, 411 HSCs per mL (7.5% of total cells) (Figure 1).

### 3.2. Motor Skills Rotarod Test

The animals were trained and tested on the Rotarod to assess their motor skills ability. The control group, treated with Lactated Ringer’s solution immediately post TBI (Group 3), saw an average decrease in motor skills ability of −40.0% ± 6.0 1-day post TBI. The group treated with SVF 3 days post TBI (Group 2) saw an average decrease in motor skills of −28% ± 11.0 as seen in Figure 2. Conversely, the group treated immediately post TBI with SVF (Group 1) saw an average improvement of 7.0% ± 9.0 1 day post TBI. By day 4, Group 3 had regressed by −50% ± 7.0. Group 2 was treated with SVF on day 3 and by day 4, Group 2 was only 2% ± 12.0 worse than the original baseline testing. Group 1 was 27% ± 12.0 above baseline on day 4. By day 7, Group 3 was −48% ± 9.0 below baseline. Group 2 was −5.0% ± 13.0 below baseline on day 7. Group 1 was 22% ± 12.0 above baseline on day 7. On the final day of testing, day 14, Group 3 was 26% ± 11.0 below baseline testing. Group 2 was 4% ± 8.0 above baseline by day 14. Group 1 finished the experiment 30% ± 7.0 above baseline testing on day 14. A repeated measures analysis showed *p* < 0.01 for this data and F = 12.572. Comparisons were made by using the Tukey HSD test which indicated an honestly significant different because of the statistically significant difference in the One-Way ANOVA for Days 1, 4, 7, and 14 respectively (SD = 0.0002, 0.0005, 0.0007, 0.0002) in all groups: between Group 1 and Group 3 (M = 0.4726, 0.772, 0.700, 0.555), Group 2 and Group 3 (M = 0.118, 0.476, 0.428, 0.298), and a small difference between Group 1 and 2 (M = 0.354, 0.295, 0.275, 0.257).

When animals were not treated with SVF immediately post TBI within the first 2 h after injury (Group 2 and 3), their rotarod performances decreased the following day by −28% to −40%. Conversely, the group treated immediately with SVF (Group 1) improved on average by 7%, showing no signs of injury. Four days post TBI, Group 3 regressed further, performing −50% worse on average. Group 2, which was treated with SVF 3 days post TBI, showed significant improvement on day 4, now only −2% worse than original baseline testing. Meanwhile, Group 1 showed continued improvement, 27% better than baseline on average. Group 3 was −48% worse than baseline on day 7 and −26% worse than baseline 14 days post TBI. Group 2 was −5% worse than baseline on day 7, but by day 14, was 4% above baseline. Group 1 continued to outperform baseline testing, 22% above on day 7 and 30% above on day 14.

### 3.3. Morris Water Maze Memory Test

The animals were trained and tested in the Morris Water Maze (MWM) test to assess and track their memories throughout the experiment. Only groups 1 and 3 were compared in this part of the experiment and were found statistically significant. One day post TBI, Group 3′s swim latency times increased 152% ± 32.0 while Group 1 actually improved their swim latency times on average by −29% ± 16.0 as seen in Figure 3. On day 4, Group 3′s swim latency times were on average 110% ± 59.0 longer than baseline while Group 1 was −52% ± 9.0 faster. Group 3 improved its average swim latency times to 59% ± 54 slower than baseline by day 7 while Group 1 was −49% ± 6.0 faster that its baseline testing. On day 14, Group 3 had recovered and its average swim latency times were −22% ± 14.0 faster than baseline. Group 1 was −63 ± 13% faster than baseline on day 14. A repeated measures ANOVA analysis for this data was statistically significant (*p* < 0.01 and F = 20.794). Comparisons were made by using the Tukey HSD test which indicated an honestly significant different because of the statistically significant difference in the One-Way ANOVA for Days 1, 4, 7, and 14 respectively (SD = 0.009, 0.0688, 0.040, 0.0005) in all groups: between Group 1 and Group 3 (M = 1.807, 1.624, 1.077, 0.409), Group 2 and Group 3 (M = 1.728, 1.410, 1.233, 0.552), and a small difference between Group 1 and 2 (M = 0.079, 0.214, 0.156, 0.143).

The Morris Water Maze test showed significant results in memory retention when Group 1 and Group 3 were compared. One day post TBI, Group 3′s swim latency times increased by 152%. The animals took much longer to find the hidden platform and showed much confusion in doing so, repeatedly swimming around the edges of the test pool. Yet one day post TBI, Group 1 decreased swim latency times by −29% on average. This trend continued through the 14-day observation period with Group 1 decreasing swim latency times by −52%, −49%, and −63% on day 4, day 7, and day 14 respectively showing no signs of memory loss from the TBI. The TBI appeared to erase Group 3′s memory, forcing the group to relearn the MWM test, as the group’s swim latency times decreased from 152% on day one, to 110%, 59% and finally −22% on day 4, day 7, and day 14 respectively. Group 3′s improvement trends are indicative of animals just learning the MWM for the first time, even though this group had been trained just as Group 1 had been trained and received the same TBI.

### 3.4. Histological Assay

Cortex tissue samples were sliced at 25 μm and are shown in Figure 4. They were then observed via fluorescence microscopy to highlight the presence of any fluorescent Q-Dot labeled cells that might be present. When tissue samples from Group 1 were reviewed, Q-Dots were readily observed in the left frontal cortex of the brain. 9 out of 10 tissue samples contained at least 1 Q-Dot labeled cell, and six of 10 contained two or more.

Q-Dots were also observed in the cortex tissue samples obtained from Group 2 animals. The Q-Dot labeled cells were less readily seen than those from Group 1, but still found in about 4 of 10 cortex tissue samples. These Q-Dot labeled cells were also more likely to be found on the periphery of the cortex associated with the TBI site, compared to those from Group 1, which were found throughout the cortex and site of injury.

In both cases, Q-Dot labeled cells were found well beyond the Pia mater and imbedded deep within the cortex of the neural tissue. The majority of Q-Dot labeled cells were found at or near the left frontal neocortex, the site of injury. Yet a few Q-Dots were observed directly opposite the site of impact in the lower right region of the amygdala.

## 4. Discussion

The Invitrogen Countess was used to obtain an approximate estimation of the cells within the SVF. Flow cytometry showed that each animal was administered approximately 5700 ADSCs and 5200 HSCs, making up 8.3% and 7.5% of the total administered cell count respectively.

All TBI were administered to the left frontal cortex of the animals’ brains, the region responsible for voluntary movement, decision-making, and even memory. For this reason, the rotarod test was used to monitor motor skills ability while the Morris Water Maze test was used to monitor memory retention. The animals were trained for three days prior to TBI. Their averaged performance numbers from the 3rd day of training were used as a baseline for analyzing their subsequent performances post TBI and treatment.

The Rotarod and MWM results suggest that SVF administered just after a TBI could mitigate, even prevent motor skills and memory deficits otherwise incurred from a TBI. Likewise, the rotarod tests also showed that an SVF treatment three days post injury could also help speed up the recovery process by 30–50%.

The histological review was conducted to see if any SVF Q-Dot labeled cells successfully migrated to the brain, specifically to the site of injury in the left frontal cortex. Q-Dot labeled cells were numerously found in the injury site of Group 1 animals. Ninety percent of Group 1 animals’ tissue samples obtained from the site of injury where the shockwave was specifically directed (left frontal cortex) showed at least one Q-Dot labeled cell, while 60% showed two or more. Q-Dot labeled cells were also observed in the injury site of Group 2 animals’ tissue samples, although less frequently around 40%. The Q-Dot labeled cells were likely more abundant in Group 1 animals due to the greater inflammation in the injury site at the time and the greater permeability of the blood brain barrier. Since Group 2 animals received SVF treatment three days post TBI, their blood brain barrier certainly had time to recover and neuro-inflammation likely decreased [15].

The Q-Dot labeled cells in Group 1 animals were found numerously throughout the injury site. They were found within the depths of the cortical neurons, beyond the Pia mater indicating they were able to migrate well beyond the blood brain barrier. The Q-Dot labeled cells in Group 2 animals were observed beyond the Pia mater as well, suggesting that they too crossed the blood brain barrier. While observed superior and inferior to the Pia mater, the cells from Group 2 were not found nearly as deep within the injury site as those found in Group 1 tissue samples.

The finding of Q-Dot labeled cells in the neural cortex suggests that the tail vein administered cells successfully migrated through the venous system to the heart, passed through the pulmonary system, cleared the capillaries of the lungs, exited the arterial system and found their way to the site of injury in the brain. Previous studies often note that IV administered stem cells become lodged in the capillaries of the lungs and are impeded from finding and treating their intended targets [16]. While Q-Dot labeled cells were found in the lungs and other filtration organs in this study, a large number successfully became implanted in the neural cortex demonstrating the efficacy of fresh SVF to home in on sites of inflammation. This study makes no determination on the viability or functionality of the SVF at the time the animals were sacrificed, rather only noting the presence of the SVF cells and the correlation to memory and motor skills.

Treatment of TBI can be a race against time. An initial TBI induces permeability of the blood brain barrier for up to four hours [15]. Group 1 animals that were treated with SVF immediately post TBI, saw a much greater number of Q-Dot labeled cells imbedded within the injury site compared to Group 2 animals. While other studies note that stem cells become lodged in the lungs [16,17,18], this could be a result of treating the cells with trypsin. It is suggested that cells treated with trypsin may become lodged in the lungs for at least 19 h [19]. This seemingly prevents the cells from reaching the brain within the 1 to 6 h window of BBB permeability [20]. Since the SVF used in this study was fresh and not treated with trypsin, and as a result of findings like Image A, the SVF most likely bypassed the lungs quickly. Although observed in fewer numbers, the histology from Group 2 animals still showed that Q-Dot labeled cells were able to bypass the BBB and become lodged in the injury site.

Another benefit of SVF and the ADSCs within it are its immunomodulatory effects. ADSCs have a greater ability to suppress the immune system compared to bone marrow derived stem cells (BMSCs) [21,22]. TBI causes an up-regulated immune response, which can have both positive and negative effects [9]. In response to TBI, neutrophils play a neuroprotective role at first, but ultimately can breakdown the BBB and promote the release of increased reactive oxygen species (ROS), causing further trauma and cell death [23]. Because of ADSCs inherent ability to suppress the immune response, the damage from neutrophils and macrophages post TBI is mitigated [9,21,22]. The superior motor skills abilities of Group 1 compared Group 3 and even Group 2, suggests that immediate SVF treatment can prevent serious damage from immune system induced ROS. The motor skills testing indicates that deployment of SVF three days post TBI can be beneficial in recovery compared to the sham treatment Group 3, but the statistically significant differential between Group 1 and Group 2 also indicates the substantial amount of damage caused by the ROS within the first three days of injury.

## 5. Conclusions

This study showed that the administration of SVF was successful in mitigating the effects of TBI in rats. Neurological damage as measured in motor skills ability and memory retention was seemingly prevented when SVF was administered immediately post TBI. Positive effects were also observed in motor skills ability when SVF was administered three days post TBI.

This study also showed that Q-Dot labeled cells within the SVF, many likely stem cells, were able to successfully migrate from the tail vein to the parenchyma of the brain injury. These cells also successfully lodged there and were found 14 days after injection.

The combination of the positive motor skills and memory test results along with the histology that revealed the presence of Q-Dots labeled cells suggests that SVF played a strong role in mitigating the effects of TBI and treating the induced injury. This study was designed to present a noninvasive TBI treatment model that could be easily applied to military personnel, football players, or any other victims of TBI.

## Figures and Tables

**Figure 1 brainsci-08-00112-f001:**
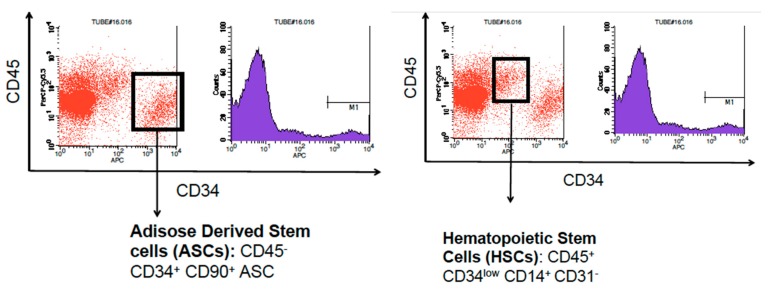
Flowcytometry analysis of stromal vascular fraction shows the presence of adipose derived stem cells characterized by CD45^-^ , CD34^+^ , CD90^+^ cell surface markers and hematopoietic stem cells characterized by CD45^+^, CD34^low^, CD14^+^, and CD31^-^ cell surface markers. ^.^

**Figure 2 brainsci-08-00112-f002:**
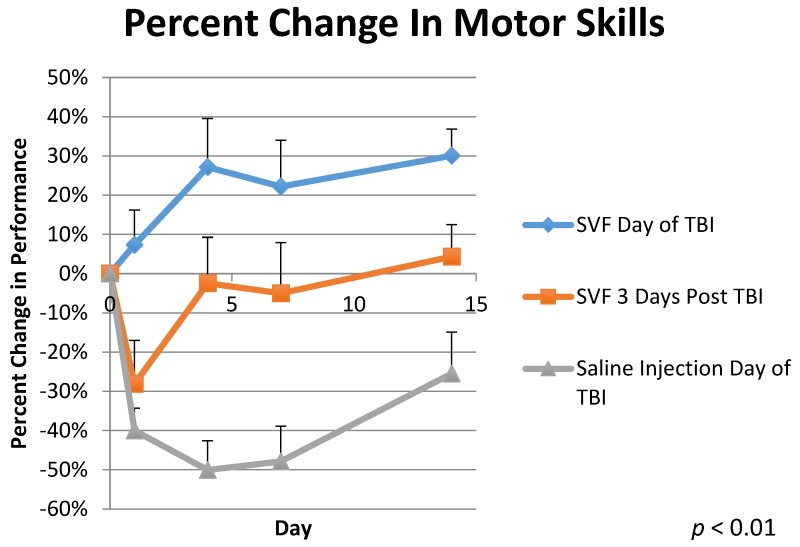
Percent Change in Motor Skills shows a comparison between three groups. Group 1 was treated with stromal vascular fraction the day of traumatic brain injury administration. Group 2 was treated with SVF 3 days post TBI, and Group 3 (control) was treated with Lactated Ringer’s solution the day of TBI. Behavioral testing was observed for 14 days post TBI. *N* = 6 for each treatment. *p* < 0.01.

**Figure 3 brainsci-08-00112-f003:**
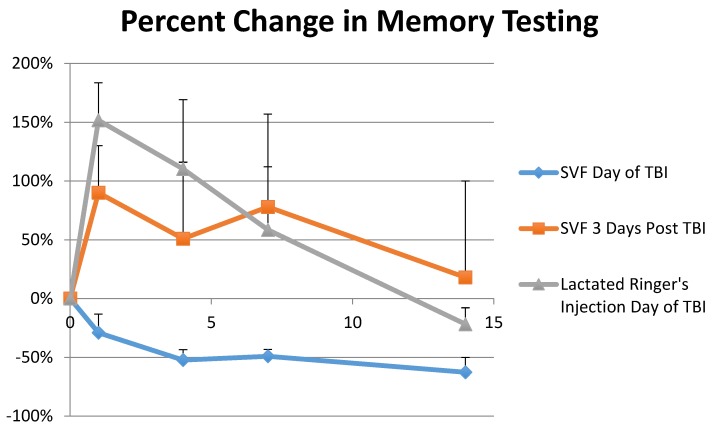
Percent Change in Memory shows a comparison between three groups: Group 1, which was treated with SVF the day of TBI and Group 3, which was treated with Lactated Ringer’s solution the day of TBI. The test was continued for 14 days. *N* = 6 per treatment. *p* < 0.01. Data from Group 2, rats treated with SVF 72 h post injury, is also included but not statistically significant.

**Figure 4 brainsci-08-00112-f004:**
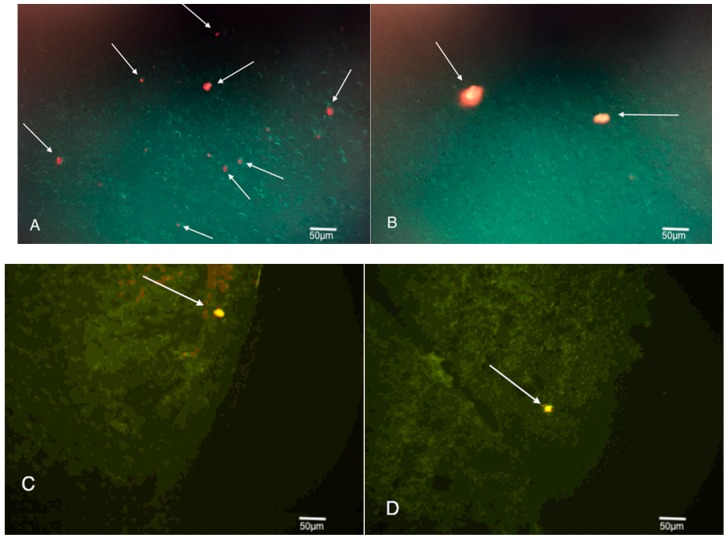
Images A and B were taken at 4× magnification and show SVF Q-Dot labeled cells within the left frontal cortex of Group 1 animals. Images C and D were taken at 4× magnification and show SVF Q-Dot labeled cells within the left frontal cortex of Group 2 animals.

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
