# Peer review of "Effective Treatment of Traumatic Brain Injury in Rowett Nude Rats with Stromal Vascular Fraction Transplantation"

_brainsci, 2018, doi:10.3390/brainsci8060112_

Round 1

Reviewer 1 Report

The authors propose that SVF may assist in post-TBI recovery.  This effort is applauded however, the experiments to determine severity of injury are too sparse to even understand what injury has been sustained.

TBI is defined as “a blow to the head causes the brain to be displaced beyond the blood brain barrier causing a subdural hematoma and damaging neuronal cells”. While this is the case in some TBIs, it is a gross and misleading understatement of TBI.  There is a broad range of injuries that are sustained and a broad range of severities with TBI.  The authors are advised to revise this statement significantly.

Again, the following statement is incorrect: “Currently, individuals are directed to rest until symptoms dissipate or go away completely [14].”  The authors should review current guidelines associated with concussion.

There is not sufficient evidence that SVF reach the brain and are functional.  The evidence provided is extremely sparse and the images are of poor quality.  Also, the validation that acoustic wave induces TBI is not sufficiently characterized.  I read the previous article (REF 19) and no information exists regarding glial activation, cellular damage etc. As it stands, more characterization is needed before this model is validated to a degree that any therapeutic intervention can be tested.

Evidence provided does not support conclusions.

Author Response

First, I want to thank you for your constructive critiques. I have responded to most of them with edits in the paper.

Additional comments were added in the introduction to explain the induction of a blast TBI, differing from a conventional TBI.

Additional statements were added about the current accepted treatment options for TBI, although minimal.

It is understood that the characterization of the shockwave model is new and has limited publications compared to other typically used TBI models. It is agreed that there is much more information that can be investigated regarding this original model. Given this, there is undoubted cellular injury that results in gross tissue damage, as well as obvious and detectable memory and motor skills deficit, making this a particularly desirable model.

There are many more studies to be done based on this initial work. But hopefully, our main findings that fresh SVF (not culture expanded) can make it to the site of injury in the brain when administered via intravenous infusion, serves as the grand finding here, evidenced by the histology. The secondary results that showed memory and motor skills improvements when SVF was administered shows that this might be a groundbreaking treatment modality and should encourage more research that can look at the viability of cells, the functionality of the cells and more. I hope this paper encourages all of that, as myself and our team continues to pursue these findings as well!

Again, I really appreciate the critical reading of this paper. Your questions are my questions as well and we will look to hopefully find the answers soon and in future studies.

Best,

Sean Berman

Reviewer 2 Report

This study investigated whether stromal vascular fraction (SVF) can assist in post-TBI recovery using acoustic waves (5.0 bar)-induced TBI in the cortex of adult Rowett Nude (RNU) rats. SVF consisting of many cells including stem cells emerges as a promising cell therapy, and it is warranted to test its efficacy in well-established TBI models. This study is original and interesting. There are several major concerns including the reproducibility of this new TBI model without full characterization, the presentation of data on motor Rotarod and Morris water maze, use of chloral hydrate without justification, tracking Q-dot labeled SVF cells in the brain, a large proportion of discussion overlapping with results. The following comments require attention.

1. Introduction. Page 1, Line 43: “to actively treat TBI in rats” is confusing. actively = acutely?

2. Methods. 2.6. Histology. The use of chloral hydrate for anesthesia or euthanasia – a compound that has been deemed generally unacceptable by international laboratory animal science/veterinary entities for these procedures–without appropriate scientific justification.

3. Methods. 2.7. Statistics. What post hoc test was applied after ANOVA to detect significance between the different groups?

4. Results. In Figure 2, Y axis label is confusing. It would be better to present the data as in their previous paper (Brain Sci. 2017, 7, 59). On Page 6, Lines 181-182: For Morris water test, a repeated measures ANOVA analysis for this data was statistically significant (p<0.01 and F= 20.794). An appropriate post-hoc test following ANOVA should be performed and reported in this study.

5. Results. 3.4. Histological Assay.

5.1 What does “the presence of any would be fluorescent Q-Dot labeled cells” mean?

5.2 Scale bars were missing in these images. 

5.3 It seems that the fluorescence detected in the brain comes from cell aggregate, especially in Images B-D. It is unclear how the authors distinguished the Q-dot labeled SVF alive cells from dead Q-dot cells or other cells that can pick up the dead Q-dot SVF cells like microglia.

5.4 There were no quantitative data on Q-dot labeled cells in the brains.

6. Results. Is there any evidence for hippocampal damage detected in this new model of TBI?

7. Discussion.

7.1 Detailed description of data in Discussion can be moved to Results section.

7.2 Page 7, Line 214: “When animals were not treated with SVF post TBI (Group 2 and 3)” is incorrect. In fact, animals in Group 2 were treated with SVF 3 days post TBI. Please clarify it.

7.3 The authors assumed that more SVF cells detected in brain from Group 1 (immediate injection post TBI) were due to greater inflammation and BBB permeability at this time point. There were no data on inflammation and BBB damage reported for this new TBI model. The time course and magnitude of inflammation and BBB damage differ in various TBI models.

7.4 There were no data on neutrophils, macrophage/microglia and ROS assay in this study to support the statement “The superior motor skills abilities of Group 1 compared Group 3 and even Group 2, suggests that immediate SVF treatment can prevent serious damage from immune system induced ROS”.

Author Response

Thank you for your critical reading of this paper. It undoubtedly has helped me to improve this work and I think you'll find that many of your critiques have been addressed in our submitted revision.

I went through and tried to address as many of your critiques as possible.

Our reference to "actively treating the rats" is in reference to the fact that almost all of the "treatment" options currently accepted and available for TBI are passive treatments based upon rest and the mitigation/suppression of symptoms, as opposed to the active treatment of the cellular injury. You are right to note that treating the acute injury did show to be more effective.

The use of chloral hydrate as an anesthetic solution was IACUC approved for this research.

F, P, F critical levels and variance degrees were calculated for each and everyone of our animal tests and those determined to be significant were noted in this study. 

Per your recommendation, the Y-axis has been changed.

The wording of "would be labeled Q-dots" has been changed.

Scale bars have been included.

The viability of the SVF was not determined in this study. The major findings showed that when the rats were given SVF via intravenous infusion, their memory and motor skills improved. We also saw that the Q-dot labeled SVF was found at the site of injury in the brain. The positive correlation between the presence of the SVF and the improved testing conditions was the significant finding. Of course, there are many future studies than can and should be done based on our novel findings.

Slight hippocampal damage was detected and commented on, but not significant.

Detailed description of discussion section for memory and motor skills results was moved to the results section.

The wording regarding which groups received treatment was revised and clarified.

The reasons why SVF improved the neurological condition of the animals and in result, their performance in memory and motor skills testing was based on related publications and included citations.

I hope these responses and the changes made in the paper are satisfactory to you and further, go to show my appreciation for your constructive critiques. I have taken much of what you have observed and used it to make this paper even stronger.

Many thanks,

-Sean Berman

Round 2

Reviewer 1 Report

the following sentence must be reworded:

TBI occurs when a blow to the head causes the brain to be displaced beyond the blood brain barrier causing a subdural hematoma and damaging neuronal cells [2]. 

TBI does not always cause a subdural hematoma.

"TBI in which the brain is displaced beyond the blood brain barrier can cause a subdural hematoma leading to neuronal cell damage."

Author Response

The statement in the introduction has been reworded. Thank you for your careful attention to detail.

Reviewer 2 Report

The authors have addressed most of comments.

Although the authors claimed that F, P, F critical levels and variance degrees were calculated, an appropriate post-hoc test used following ANOVA should be reported in this manuscript.

Please complete the Section of Author Contributions for each author (not just XX, YY, WW).

A minor typo:  Page 1, Line 42: to mitigate paid =to mitigate pain?

Author Response

Author contributions and the small typo have been changed.

Post hoc Tukey HSD tests were run and reported on.

Thank you for your attention to detail and constructive observations.

Many Thanks!